# Predictive PAC Learning and Process Decompositions

**Cosma Rohilla Shalizi**
Statistics Department
Carnegie Mellon University
Pittsburgh, PA 15213 USA
cshalizi@cmu.edu

**Aryeh Kontorovich**
Computer Science Department
Ben Gurion University
Beer Sheva 84105 Israel
karyeh@cs.bgu.ac.il

## Abstract

We informally call a stochastic process learnable if it admits a generalization error approaching zero in probability for any concept class with finite VC-dimension (IID processes are the simplest example). A mixture of learnable processes need not be learnable itself, and certainly its generalization error need not decay at the same rate. In this paper, we argue that it is natural in predictive PAC to condition not on the past observations but on the mixture component of the sample path. This definition not only matches what a realistic learner might demand, but also allows us to sidestep several otherwise grave problems in learning from dependent data. In particular, we give a novel PAC generalization bound for mixtures of learnable processes with a generalization error that is not worse than that of each mixture component. We also provide a characterization of mixtures of absolutely regular ($\beta$-mixing) processes, of independent probability-theoretic interest.

## 1 Introduction

Statistical learning theory, especially the theory of "probably approximately correct" (PAC) learning, has mostly developed under the assumption that data are independent and identically distributed (IID) samples from a fixed, though perhaps adversarially-chosen, distribution. As recently as 1997, Vidyasagar [1] named extending learning theory to stochastic processes of dependent variables as a major open problem. Since then, considerable progress has been made for specific classes of processes, particularly strongly-mixing sequences and exchangeable sequences. (Especially relevant contributions, for our purposes, came from [2, 3] on exchangeability, from [4, 5] on absolute regularity[1], and from [6, 7] on ergodicity; others include [8, 9, 10, 11, 12, 13, 14, 15, 16, 17].) Our goals in this paper are to point out that many practically-important classes of stochastic processes possess a special sort of structure, namely they are convex combinations of simpler, extremal processes. This both demands something of a re-orientation of the goals of learning, and makes the new goals vastly easier to attain than they might seem.

**Main results**   Our main contribution is threefold: a conceptual definition of learning from non-IID data (Definition 1) and a technical result establishing tight generalization bounds for mixtures of learnable processes (Theorem 2), with a direct corollary about exchangeable sequences (Corollary 1), and an application to mixtures of absolutely regular sequences, for which we provide a new characterization.

**Notation**   $X_1, X_2, \ldots$ will be a sequence of dependent random variables taking values in a common measurable space $\mathcal{X}$, which we assume to be "standard" [18, ch. 3] to avoid technicalities, implying

that their $\sigma$-field has a countable generating basis; the reader will lose little if they think of $\mathcal{X}$ as $\mathbb{R}^d$. (We believe our ideas apply to stochastic processes with multidimensional index sets as well, but use sequences here.) $X_i^j$ will stand for the block $(X_i, X_{i+1}, \ldots X_{j-1}, X_j)$. Generic infinite-dimensional distributions, measures on $\mathcal{X}^\infty$, will be $\mu$ or $\rho$; these are probability laws for $X_1^\infty$. Any such stochastic process can be represented through the shift map $\tau : \mathcal{X}^\infty \mapsto \mathcal{X}^\infty$ (which just drops the first coordinate, $(\tau x)_t = x_{t+1}$), and a suitable distribution of initial conditions. When we speak of a set being invariant, we mean invariant under the shift. The collection of all such probability measures is itself a measurable space, and a generic measure there will be $\pi$.

## 2  Process Decompositions

Since the time of de Finetti and von Neumann, an important theme of the theory of stochastic processes has been finding ways of representing complicated but structured processes, obeying certain symmetries, as mixtures of simpler processes with the same symmetries, as well as (typically) some sort of 0-1 law. (See, for instance, the beautiful paper by Dynkin [19], and the statistically-motivated [20].) In von Neumann's original ergodic decomposition [18, §7.9], stationary processes, whose distributions are invariant over time, proved to be convex combinations of stationary *ergodic* measures, ones where all invariant sets have either probability 0 or probability 1. In de Finetti's theorem [21, ch. 1], exchangeable sequences, which are invariant under permutation, are expressed as mixtures of IID sequences[2]. Similar results are now also known for asymptotically mean stationary sequences [18, §8.4], for partially-exchangeable sequences [22], for stationary random fields, and even for infinite exchangeable arrays (including networks and graphs) [21, ch. 7].

The common structure shared by these decompositions is as follows.

1. The probability law $\rho$ of the composite but symmetric process is a convex combination of the simpler, extremal processes $\mu \in \mathcal{M}$ with the same symmetry. The infinite-dimensional distribution of these extremal processes are, naturally, mutually singular.

2. Sample paths from the composite process are generated hierarchically, first by picking an extremal process $\mu$ from $\mathcal{M}$ according to a some measure $\pi$ supported on $\mathcal{M}$, and then generating a sample path from $\mu$. Symbolically,

$$
\begin{aligned}
\mu &\sim \pi \\
X_1^\infty \mid \mu &\sim \mu
\end{aligned}
$$

3. Each realization of the composite process therefore gives information about only a single extremal process, as this is an invariant along each sample path.

## 3  Predictive PAC

These points raise subtle but key issues for PAC learning theory. Recall the IID case: random variables $X_1, X_2, \ldots$ are all generated from a common distribution $\mu^{(1)}$, leading to an infinite-dimensional process distribution $\mu$. Against this, we have a class $\mathcal{F}$ of functions $f$. The goal in PAC theory is to find a sample complexity function[3] $s(\epsilon, \delta, \mathcal{F}, \mu)$ such that, whenever $n \geq s$,

$$
\mathbb{P}_\mu \left( \sup_{f \in \mathcal{F}} \left| \frac{1}{n} \sum_{t=1}^{n} f(X_t) - \mathbf{E}_\mu[f] \right| \geq \epsilon \right) \leq \delta \tag{1}
$$

That is, PAC theory seeks finite-sample uniform law of large numbers for $\mathcal{F}$.

Because of its importance, it will be convenient to abbreviate the supremum,

$$
\sup_{f \in \mathcal{F}} \left| \frac{1}{n} \sum_{t=1}^{n} f(X_t) - \mathbf{E}_\mu[f] \right| \equiv \Gamma_n
$$

using the letter "$\Gamma$" as a reminder that when this goes to zero, $\mathcal{F}$ is a Glivenko-Cantelli class (for $\mu$). $\Gamma_n$ is also a function of $\mathcal{F}$ and of $\mu$, but we suppress this in the notation for brevity. We will also pass over the important and intricate, but fundamentally technical, issue of establishing that $\Gamma_n$ is measurable (see [23] for a thorough treatment of this topic).

What one has in mind, of course, is that there is a space $\mathcal{H}$ of predictive models (classifiers, regressions, ...) $h$, and that $\mathcal{F}$ is the image of $\mathcal{H}$ through an appropriate loss function $\ell$, i.e., each $f \in \mathcal{F}$ can be written as

$$f(x) = \ell(x, h(x))$$

for some $h \in \mathcal{H}$. If $\Gamma_n \to 0$ in probability for this "loss function" class, then empirical risk minimization is reliable. That is, the function $\hat{f}_n$ which minimizes the empirical risk $n^{-1} \sum_t f(X_t)$ has an expected risk in the future which is close to the best attainable risk over all of $\mathcal{F}$, $R(\mathcal{F}, \mu) = \inf_{f \in \mathcal{F}} \mathbf{E}_\mu [f]$. Indeed, since when $n \geq s$, with high ($\geq 1 - \delta$) probability all functions have empirical risks within $\epsilon$ of their true risks, with high probability the true risk $\mathbf{E}_\mu \left[ \hat{f}_n \right]$ is within $2\epsilon$ of $R(\mathcal{F}, \mu)$. Although empirical risk minimization is not the only conceivable learning strategy, it is, in a sense, a canonical one (computational considerations aside). The latter is an immediate consequence of the VC-dimension characterization of PAC learnability:

**Theorem 1** *Suppose that the concept class $\mathcal{H}$ is PAC learnable from IID samples. Then $\mathcal{H}$ is learnable via empirical risk minimization.*

PROOF: Since $\mathcal{H}$ is PAC-learnable, it must necessarily have a finite VC-dimension [24]. But for finite-dimensional $\mathcal{H}$ and IID samples, $\Gamma_n \to 0$ in probability (see [25] for a simple proof). This implies that the empirical risk minimizer is a PAC learner for $\mathcal{H}$. $\square$

In extending these ideas to non-IID processes, a subtle issue arises, concerning which expectation value we would like empirical means to converge towards. In the IID case, because $\mu$ is simply the infinite product of $\mu^{(1)}$ and $f$ is a function on $\mathcal{X}$, we can without trouble identify expectations under the two measures with each other, and with expectations conditional on the first $n$ observations:

$$\mathbf{E}_\mu [f(X)] = \mathbf{E}_{\mu^{(1)}} [f(X)] = \mathbf{E}_\mu [f(X_{n+1}) \mid X_1^n]$$

Things are not so tidy when $\mu$ is the law of a dependent stochastic process.

In introducing a notion of "predictive PAC learning", Pestov [3], like Berti and Rigo [2] earlier, proposes that the target should be the conditional expectation, in our notation $\mathbf{E}_\mu [f(X_{n+1}) \mid X_1^n]$. This however presents two significant problems. First, in general there is no single value for this — it truly is a function of the past $X_1^n$, or at least some part of it. (Consider even the case of a binary Markov chain.) The other, and related, problem with this idea of predictive PAC is that it presents learning with a perpetually moving target. Whether the function which minimizes the empirical risk is going to do well by this criterion involves rather arbitrary details of the process. To truly pursue this approach, one would have to actually learn the predictive dependence structure of the process, a quite formidable undertaking, though perhaps not hopeless [26].

Both of these complications are exacerbated when the process producing the data is actually a mixture over simpler processes, as we have seen is very common in interesting applied settings. This is because, in addition to whatever dependence may be present within each extremal process, $X_1^n$ gives (partial) information about what that process is. Finding $\mathbf{E}_\rho [X_{n+1} \mid X_1^n]$ amounts to first finding all the individual conditional expectations, $\mathbf{E}_\mu [X_{n+1} \mid X_1^n]$, and then averaging them with respect to the posterior distribution $\pi(\mu \mid X_1^n)$. This averaging over the posterior produces additional dependence between past and future. (See [27] on quantifying how much extra apparent Shannon information this creates.) As we show in §4 below, continuous mixtures of absolutely regular processes are far from being absolutely regular themselves. This makes it exceedingly hard, if not impossible, to use a single sample path, no matter how long, to learn anything about global expectations.

These difficulties all simply dissolve if we change the target distribution. What a learner should care about is not averaging over some hypothetical prior distribution of inaccessible alternative dynamical systems, but rather what will happen in the future of the particular realization which provided the training data, and must continue to provide the testing data. To get a sense of what this is means,

notice that for an ergodic $\mu$,

$$\mathbf{E}_\mu [f] = \lim_{m \to \infty} \frac{1}{m} \sum_{i=1}^{m} \mathbf{E} [f(X_{n+i}) \mid X_1^n]$$

(from [28, Cor. 4.4.1]). That is, matching expectations under the process measure $\mu$ means controlling the *long-run average* behavior, and not just the one-step-ahead expectation suggested in [3, 2]. Empirical risk minimization now makes sense: it is attempting to find a model which will work well not just at the next step (which may be inherently unstable), but will continue to work well, on average, indefinitely far into the future.

We are thus led to the following definition.

**Definition 1** *Let $X_1^\infty$ be a stochastic process with law $\mu$, and let $\mathcal{I}$ be the $\sigma$-field generated by all the invariant events. We say that $\mu$ admits* predictive PAC learning *of a function class $\mathcal{F}$ when there exists a sample-complexity function $s(\epsilon, \delta, \mathcal{F}, \mu)$ such that, if $n \geq s$, then*

$$\mathbb{P}_\mu \left( \sup_{f \in \mathcal{F}} \left| \frac{1}{n} \sum_{t=1}^{n} f(X_t) - \mathbf{E}_\mu [f | \mathcal{I}] \right| \geq \epsilon \right) \leq \delta$$

*A class of processes $\mathcal{P}$ admits of* distribution-free predictive PAC learning *if there exists a common sample-complexity function for all $\mu \in \mathcal{P}$, in which case we write $s(\epsilon, \delta, \mathcal{F}, \mu) = s(\epsilon, \delta, \mathcal{F}, \mathcal{P})$.*

As is well-known, distribution-free predictive PAC learning, in this sense, is possible for IID processes ($\mathcal{F}$ must have finite VC dimension). For an ergodic $\mu$, [6] shows that $s(\epsilon, \delta, \mathcal{F}, \mu)$ exist and is finite if and only, once again, $\mathcal{F}$ has a finite VC dimension; this implies predictive PAC learning, but not distribution-free predictive PAC. Since ergodic processes can converge arbitrarily slowly, some extra condition must be imposed on $\mathcal{P}$ to ensure that dependence decays fast enough for each $\mu$. A sufficient restriction is that all processes in $\mathcal{P}$ be stationary and absolutely regular ($\beta$-mixing), with a common upper bound on the $\beta$ dependence coefficients, as [5, 14] show how to turn algorithms which are PAC on IID data into ones where are PAC on such sequences, with a penalty in extra sample complexity depending on $\mu$ only through the rate of decay of correlations[4]. We may apply these familiar results straightforwardly, because, when $\mu$ is ergodic, all invariant sets have either measure 0 or measure 1, conditioning on $\mathcal{I}$ has no effect, and $\mathbf{E}_\mu [f \mid \mathcal{I}] = \mathbf{E}_\mu [f]$.

Our central result is now almost obvious.

**Theorem 2** *Suppose that distribution-free prediction PAC learning is possible for a class of functions $\mathcal{F}$ and a class of processes $\mathcal{M}$, with sample-complexity function $s(\epsilon, \delta, \mathcal{F}, \mathcal{P})$. Then the class of processes $\mathcal{P}$ formed by taking convex mixtures from $\mathcal{M}$ admits distribution-free PAC learning with the same sample complexity function.*

PROOF: Suppose that $n \geq s(\epsilon, \delta, \mathcal{F})$. Then by the law of total expectation,

$$\begin{align}
\mathbb{P}_\rho (\Gamma_n \geq \epsilon) &= \mathbf{E}_\rho [\mathbb{P}_\rho (\Gamma_n \geq \epsilon \mid \mu)] \tag{2} \\
&= \mathbf{E}_\rho [\mathbb{P}_\mu (\Gamma_n \geq \epsilon)] \tag{3} \\
&\leq \mathbf{E}_\rho [\delta] = \delta \tag{4}
\end{align}$$

□

In words, if the same bound holds for each component of the mixture, then it still holds after averaging over the mixture. It is important here that we are only attempting to predict the long-run average risk along the continuation of the same sample path as that which provided the training data; with this as our goal, almost all sample paths looks like — indeed, *are* — realizations of single components of the mixture, and so the bound for extremal processes applies directly to them[5]. By contrast, there may be no distribution-free bounds at all if one does not condition on $\mathcal{I}$.

A useful consequence of this innocent-looking result is that it lets us *immediately* apply PAC learning results for extremal processes to composite processes, without any penalty in the sample complexity. For instance, we have the following corollary:

**Corollary 1** *Let $\mathcal{F}$ have finite VC dimension, and have distribution-free sample complexity $s(\epsilon, \delta, \mathcal{F}, \mathcal{M})$ for all IID measures $\mu \in \mathcal{P}$. Then the class $\mathcal{M}$ of exchangeable processes composed from $\mathcal{P}$ admit distribution-free PAC learning with the same sample complexity,*

$$s(\epsilon, \delta, \mathcal{F}, \mathcal{P}) = s(\epsilon, \delta, \mathcal{F}, \mathcal{M})$$

This is in contrast with, for instance, the results obtained by [2, 3], which both go from IID sequences (laws in $\mathcal{P}$) to exchangeable ones (laws in $\mathcal{M}$) at the cost of considerably increased sample complexity. The easiest point of comparison is with Theorem 4.2 of [3], which, in our notation, shows that

$$s(\epsilon, \delta, \mathcal{M}) \leq s(\epsilon/2, \delta\epsilon, \mathcal{P})$$

That we pay no such penalty in sample complexity because our target of learning is $\mathbf{E}_\mu [f \mid \mathcal{I}]$, not $\mathbf{E}_\rho [f \mid X_1^n]$. This means we do not have to use data points to narrow the posterior distribution $\pi(\mu \mid X_1^n)$, and that we can give a much more direct argument.

In [3], Pestov asks whether "one [can] maintain the initial sample complexity" when going from IID to exchangeable sequences; the answer is yes, *if* one picks the right target. This holds whenever the data-generating process is a mixture of extremal processes for which learning is possible. A particularly important special case of this are the absolutely regular processes.

## 4   Mixtures of Absolutely Regular Processes

Roughly speaking, an **absolutely regular** process is one which is asymptotically independent in a particular sense, where the joint distribution between past and future events approaches, in $L_1$, the product of the marginal distributions as the time-lag between past and future grows. These are particularly important for PAC learning, since much of the existing IID learning theory translates directly to this setting.

To be precise, let $X_{-\infty}^\infty$ be a two-sided[6] stationary process. The $\beta$-dependence coefficient at lag $k$ is

$$\beta(k) \equiv \left\| P_{-\infty}^0 \otimes P_k^\infty - P_{-(1:k)} \right\|_{\text{TV}} \tag{5}$$

where $P_{-(1:k)}$ is the joint distribution of $X_{-\infty}^0$ and $X_k^\infty$, the total variation distance between the actual joint distribution of the past and future, and the product of their marginals. Equivalently [31, 32]

$$\beta(k) = \mathbf{E} \left[ \sup_{B \in \sigma(X_k^\infty)} \mathbb{P}\left(B \mid X_{-\infty}^0\right) - \mathbb{P}(B) \right] \tag{6}$$

which, roughly, is the expected total variation distance between the marginal distribution of the future and its distribution conditional on the past.

As is well known, $\beta(k)$ is non-increasing in $k$, and of course $\geq 0$, so $\beta(k)$ must have a limit as $k \to \infty$; it will be convenient to abbreviate this as $\beta(\infty)$. When $\beta(\infty) = 0$, the process is said to be beta mixing, or absolutely regular. All absolutely regular processes are also ergodic [32].

The importance of absolutely regular processes for learning comes essentially from a result due to Yu [4]. Let $X_1^n$ be a part of a sample path from an absolutely regular process $\mu$, whose dependence coefficients are $\beta(k)$. Fix integers $m$ and $a$ such that $2ma = n$, so that the sequence is divided into $2m$ contiguous blocks of length $a$, and define $\mu^{(m,a)}$ to be distribution of $m$ length-$a$ blocks. (That is, $\mu^{(m,a)}$ approximates $\mu$ by IID blocks.) Then $|\mu(C) - \mu^{(m,a)}(C)| \leq (m-1)\beta(a)$ [4, Lemma 4.1]. Since in particular the event $C$ could be taken to be $\{\Gamma_n > \epsilon\}$, this approximation result allows distribution-free learning bounds for IID processes to translate directly into distribution-free learning bounds for absolutely regular processes with bounded $\beta$ coefficients.

If $\mathcal{M}$ contains only absolutely regular processes, then a measure $\pi$ on $\mathcal{M}$ creates a $\rho$ which is a mixture of absolutely regular processes, or a MAR process. It is easy to see that absolute regularity of the component processes ($\beta_\mu(k) \to 0$) does not imply absolute regularity of the mixture processes ($\beta_\rho(k) \not\to 0$). To see this, suppose that $\mathcal{M}$ consists of two processes $\mu_0$, which puts unit probability mass on the sequence of all zeros, and $\mu_1$, which puts unit probability on the sequence of all ones; and that $\pi$ gives these two equal probability. Then $\beta_{\mu_i}(k) = 0$ for both $i$, but the past and the future of $\rho$ are not independent of each other.

More generally, suppose $\pi$ is supported on just two absolutely regular processes, $\mu$ and $\mu'$. For each such $\mu$, there exists a set of typical sequences $T_\mu \subset \mathcal{X}^\infty$, in which the infinite sample path of $\mu$ lies almost surely[7], and these sets do not overlap[8], $T_\mu \cap T_{\mu'} = \emptyset$. This implies that $\rho(T_\mu) = \pi(\mu)$, but

$$\rho(T_\mu \mid X^0_{-\infty}) = \begin{cases} 1 & X^0_{-\infty} \in T_\mu \\ 0 & \text{otherwise} \end{cases}$$

Thus the change in probability of $T_\mu$ due to conditioning on the past is $\pi(\mu_1)$ if the selected component was $\mu_2$, and $1 - \pi(\mu_1) = \pi(\mu_2)$ if the selected component is $\mu_1$. We can reason in parallel for $T_{\mu_2}$, and so the average change in probability is

$$\pi_1(1 - \pi_1) + \pi_2(1 - \pi_2) = 2\pi_1(1 - \pi_1)$$

and this must be $\beta_\rho(\infty)$. Similar reasoning when $\pi$ is supported on $q$ extremal processes shows

$$\beta_\rho(\infty) = \sum_{i=1}^{q} \pi_i(1 - \pi_i)$$

so that the general case is

$$\beta_\rho(\infty) = \int [1 - \pi(\{\mu\})] d\pi(\mu)$$

This implies that if $\pi$ has no atoms, $\beta_\rho(\infty) = 1$ always. Since $\beta_\rho(k)$ is non-increasing, $\beta_\rho(k) = 1$ for all $k$, for a continuous mixture of absolutely regular processes. Conceptually, this arises because of the use of infinite-dimensional distributions for both past and future in the definition of the $\beta$-dependence coefficient. Having seen an infinite past is sufficient, for an ergodic process, to identify the process, and of course the future must be a continuation of this past.

MARs thus display a rather odd separation between the properties of individual sample paths, which approach independence asymptotically in time, and the ensemble-level behavior, where there is ineradicable dependence, and indeed maximal dependence for continuous mixtures. Any one realization of a MAR, no matter how long, is indistinguishable from a realization of an absolutely regular process which is a component of the mixture. The distinction between a MAR and a single absolutely regular process only becomes apparent at the level of ensembles of paths.

It is desirable to characterize MARs. They are stationary, but non-ergodic and have non-vanishing $\beta(\infty)$. However, this is not sufficient to characterize them. Bernoulli shifts are stationary and ergodic, but not absolutely regular[9]. It follows that a mixture of Bernoulli shifts will be stationary, and by the preceding argument will have positive $\beta(\infty)$, but will not be a MAR.

Roughly speaking, given the infinite past of a MAR, events in the future become asymptotically independent as the separation between them increases[10]. A more precise statement needs to control the approach to independence of the component processes in a MAR. We say that $\rho$ is a $\tilde{\beta}$-**uniform MAR** when $\rho$ is a MAR, and, for $\pi$-almost-all $\mu$, $\beta_\mu(k) \leq \tilde{\beta}(k)$, with $\tilde{\beta}(k) \to 0$. Then if we condition on finite histories $X^0_{-n}$ and let $n$ grow, widely separated future events become asymptotically independent.

**Theorem 3** *A stationary process $\rho$ is a $\tilde{\beta}$-uniform MAR if and only if*

$$\lim_{k\to\infty}\lim_{n\to\infty}\mathbf{E}\left[\sup_{l}\sup_{B\in\sigma(X_{k+l}^{\infty})}\rho(B\mid X_1^l, X_{-n}^0) - \rho(B\mid X_{-n}^0)\right] = 0 \tag{7}$$

Before proceeding to the proof, it is worth remarking on the order of the limits: for finite $n$, conditioning on $X_{-n}^0$ still gives a MAR, not a single (albeit random) absolutely-regular process. Hence the $k\to\infty$ limit for fixed $n$ would always be positive, and indeed 1 for a continuous $\pi$.

PROOF "Only if": Re-write Eq. 7, expressing $\rho$ in terms of $\pi$ and the extremal processes:

$$\lim_{k\to\infty}\lim_{n\to\infty}\mathbf{E}\left[\sup_{l}\sup_{B\in\sigma(X_{k+l}^{\infty})}\int \left(\mu(B\mid X_1^l, X_{-n}^0) - \mu(B\mid X_{-n}^0)\right) d\pi(\mu\mid X_{-n}^0)\right]$$

As $n\to\infty$, $\mu(B\mid X_{-n}^0) \to \mu(B\mid X_{-\infty}^0)$, and $\mu(B\mid X_1^l, X_{-n}^0) \to \mu(B\mid X_{-\infty}^l)$. But, in expectation, both of these are within $\tilde{\beta}(k)$ of $\mu(B)$, and so within $2\tilde{\beta}(k)$. "If": Consider the contrapositive. If $\rho$ is not a uniform MAR, either it is a non-uniform MAR, or it is not a MAR at all. If it is not a uniform MAR, then no matter what function $\tilde{\beta}(k)$ tending to zero we propose, the set of $\mu$ with $\beta_\mu \geq \tilde{\beta}$ must have positive $\pi$ measure, i.e., a positive-measure set of processes must converge arbitrarily slowly. Therefore there must exist a set $B$ (or a sequence of such sets) witnessing this arbitrarily slow convergence, and hence the limit in Eq. 7 will be strictly positive. If $\rho$ is not a MAR at all, we know from the ergodic decomposition of stationary processes that it must be a mixture of ergodic processes, and so it must give positive $\pi$ weight to processes which are not absolutely regular at all, i.e., $\mu$ for which $\beta_\mu(\infty) > 0$. The witnessing events $B$ for these processes *a fortiori* drive the limit in Eq. 7 above zero. $\square$

## 5  Discussion and future work

We have shown that with the right conditioning, a natural and useful notion of predictive PAC emerges. This notion is natural in the sense that for a learner sampling from a mixture of ergodic processes, the only thing that matters is the future behavior of the component he is "stuck" in — and certainly not the process average over all the components. This insight enables us to adapt the recent PAC bounds for mixing processes to mixtures of such processes. An interesting question then is to characterize those processes that are convex mixtures of a given kind of ergodic process (de Finetti's theorem was the first such characterization). In this paper, we have addressed this question for mixtures of uniformly absolutely regular processes. Another fascinating question is how to extend predictive PAC to real-valued functions [33, 34].

## Footnotes

[1]Absolutely regular processes are ones where the joint distribution of past and future events approaches independence, in $L_1$, as the separation between events goes to infinity; see §4 below for a precise statement and extensive discussion. Absolutely regular sequences are now more commonly called "$\beta$-mixing", but we use the older name to avoid confusion with the other sort of "mixing".

[2]This is actually a special case of the ergodic decomposition [21, pp. 25–26].

[3]Standard PAC is defined as distribution-free, but here we maintain the dependence on $\mu$ for consistency with future notation.

[4]We suspect that similar results could be derived for many of the weak dependence conditions of [29].

[5]After formulating this idea, we came across a remarkable paper by Wiener [30], where he presents a qualitative version of highly similar considerations, using the ergodic decomposition to argue that a full dynamical model of the weather is neither necessary nor even helpful for meteorological forecasting. The same paper also lays out the idea of sensitive dependence on initial conditions, and the kernel trick of turning nonlinear problems into linear ones by projecting into infinite-dimensional feature spaces.

[6]We have worked with one-sided processes so far, but the devices for moving between the two representations are standard, and this definition is more easily stated in its two-sided version.

[7]Since $\mathcal{X}$ is "standard", so is $\mathcal{X}^\infty$, and the latter's $\sigma$-field $\sigma(X^\infty_{-\infty})$ has a countable generating basis, say $\mathcal{B}$. For each $B \in \mathcal{B}$, the set $T_{\mu,B} = \{x \in \mathcal{X}^\infty : \lim_{n \to \infty} n^{-1} \sum_{t=0}^{n-1} \mathbf{1}_B(\tau^t x) = \mu(B)\}$ exists, is measurable, is dynamically invariant, and, by Bikrhoff's ergodic theorem, $\mu(T_{\mu,B}) = 1$ [18, §7.9]. Then $T_\mu \equiv \bigcap_{B \in \mathcal{B}} T_{\mu,B}$ also has $\mu$-measure 1, because $\mathcal{B}$ is countable.

[8]Since $\mu \neq \mu'$, there exists at least one set $C$ with $\mu(C) \neq \mu'(C)$. The set $T_{\mu,C}$ then cannot overlap at all with the set $T_{\mu',C}$, and so $T_\mu$ cannot intersect $T_{\mu'}$.

[9]They are, however, mixing in the sense of ergodic theory [28].

[10]$\rho$-almost-surely, $X^0_{-\infty}$ belongs to the typical set of a unique absolutely regular process, say $\mu$, and so the posterior concentrates on that $\mu$, $\pi(\cdot \mid X^0_{-\infty}) = \delta_\mu$. Hence $\rho(X^l_1, X^\infty_{l+k} \mid X^0_{-\infty}) = \mu(X^l_{-\infty}, X^\infty_{l+k})$, which tends to $\mu(X^l_{-\infty})\mu(X^\infty_{l+k})$ as $k \to \infty$ because $\mu$ is absolutely regular.

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
