[Reviews · NeurIPS 2013]

Submitted by Assigned_Reviewer_4

This paper focuses on learning with dependent observations. Under various assumptions, it is possible to upper-bound the deviations of the empirical mean with respect to the expectation, thus allowing to study empirical risk minimization. These assumptions ranges from various mixing assumptions to so-called weak-dependence condition.

The authors remark that when P and Q are two probability distributions for the stochastic process X, when X is mixing under P and under Q, it is generally not mixing under aP + (1-a)Q. However, if E is a deviation undersirable event, and if P(E)\leq e and Q(E)\leq e, then (aP+(1-a)Q)(E)\leq e. So, it is possible to extend deviation inequalities to a wider class of probability distribution (that is, mixture of mixing disitributions). This remark might seem trivial, however, it allows some interesting generalization. For example, as exchangeable probability distributions can be written as mixtures of iid distributions, this remark extends results from iid setting to exhangeable distribution at no cost.

However:
1) to my opinion, this is a small increment to learning with dependent observations.
2) the paper is not well written. Unnecessary discussions hide the main message of the paper. On the other hand, Definition 1 is non standard and should be explained in more details. The bibliography is incomplete. A lot has been done on learning with dependent observations in Modha and Masry (IEEE Trans. Info. Theory, 1998), Meir (JMLR, 2000), a series of papers by Steinwart, e.g. Steinwart, Hush and Scovel (Journal of Multivariate Analysis, 2009), Alquier and Wintenberger (Bernoulli, 2012) among others.
Summary: An interesting remark, however:
1) this is a small increment to learning with dependent observations.
2) the paper is not well written.

Submitted by Assigned_Reviewer_5

In this paper, a PAC bound is given for mixtures of processes satisfying mixing conditions, and
a similar result is shown to exist for exchangeable sequences.

I really like this paper.
It's interesting, well-written and theoretically sound.
I personally find the introduction a bit too short;
perhaps the authors could elaborate on the main results in the intro.

How would the results of this paper generalize to the case of ergodic processes that do not necessarily satisfy any mixing conditions?

Minor comments:
I didn't understand what was meant by "of independent interest" in the last sentence of the abstract.
pg 4, second to last paragraph: "by contrast" -> By contrast
Summary: The paper is interesting, nicely written and theoretically sound.

Submitted by Assigned_Reviewer_7

The paper discusses an extension of generalization error results (more precisely, the GC type property) beyond i.i.d. data. The authors argue that one of the difficulties faced by previous attempts is in the goal itself: one should not consider the average of conditional expectations but rather the "ergodic counterpart" to the expectation.

This paper was a pleasure to read. I enjoyed the motivation and the argument in favor of the "ergodic" definition. There are no breakthroughs in this paper, but I would encourage its publication: it sets the stage for further work on non-i.i.d. extensions. The simple observation that one does not pay for the mixture of learnable processes and simply inherits their sample complexity -- is nice.


Minor:

* One missing reference that comes to mind is " Extension of the PAC Framework to Finite and Countable Markov Chains" by Gamarnik.

* line 153: beta(k) seems to be undefined at this point in the paper.
* line 180: "impose" -> "imposed"
* line 206: "by" -> "By"

Summary: This is a very well written paper that brings out many nice connections and provides an outlook on the problem formulation for learning with non i.i.d sequences. No technical breakthroughs, but a nice exposition.
Author Feedback

Author rebuttal: We thank the three referees for their helpful comments, and intend to fully address them in the revised version. In particular:

- We will elaborate upon Definition 1. In particular, we will clarify how the discussion on pp. 3--4, from line 130 onwards, serves to motivate the definition.
- We will expand the bibliography to include Modha and Masry 1998, Meir 2000, the papers by Steinwart et al, Alquier and Wintenberger 2012, and Gamarnik 2003 -- as well as placing the present result in the context of the existing ones, which will serve to underscore the novelty of our model.
- We will explain what is meant by "of independent interest" in the last sentence of the abstract (i.e., it is of interest to probability theorists, independent of learning applications).
- We will correct the few typos pointed out.